Characterization of PYL gene family and identification of HaPYL genes response to drought and salt stress in sunflower

Wang Zhaoping
Zhou Jiayan
Zou Jian
Yang Jun yangjun@cwnu.edu.cn
Chen Weiying chenweiying@cwnu.edu.cn
China West Normal University, College of Life Sciences , Nanchong , Sichuan , China
Kumar Ravinder
Electronic publication date: 2024 Mar 7
Publication date: 2024
Volume: 12
Electronic Location ID: e16831
Received 2023 Jul 3; Accepted 2024 Jan 4
Copyright: ©2024 Wang et al.
Copyright year: 2024
Copyright holder: Wang et al.
License: This is an open access article distributed under the terms of the Creative Commons Attribution License, which permits unrestricted use, distribution, reproduction and adaptation in any medium and for any purpose provided that it is properly attributed. For attribution, the original author(s), title, publication source (PeerJ) and either DOI or URL of the article must be cited.
License URL: https://creativecommons.org/licenses/by/4.0/

Keywords: Abscisic acid, Sunflower (Helianthus annuus L.), PYL, PEG6000, Salinity

Funding: Meritocracy Research Funds of China West Normal University 17YC321 Natural Science Foundation of Sichuan Province of China 2022NSFSC0163 Natural Science Foundation of Sichuan Province of China 2023NSFSC0226 National General Incubation Program of China West Normal University 19B035 Scientific Research Foundation for Doctors of China West Normal University 18Q039 This work was supported by the Meritocracy Research Funds of China West Normal University (17YC321), the Natural Science Foundation of Sichuan Province of China (2022NSFSC0163), the Natural Science Foundation of Sichuan Province of China (2023NSFSC0226), the National General Incubation Program of China West Normal University (19B035) and the Scientific Research Foundation for Doctors of China West Normal University (18Q039). The funders had no role in study design, data collection and analysis, decision to publish, or preparation of the manuscript.

==============================
In the context of global climate change, drought and soil salinity are some of the most devastating abiotic stresses affecting agriculture today. PYL proteins are essential components of abscisic acid (ABA) signaling and play critical roles in responding to abiotic stressors, including drought and salt stress. Although PYL genes have been studied in many species, their roles in responding to abiotic stress are still unclear in the sunflower. In this study, 19 HaPYL genes, distributed on 15 of 17 chromosomes, were identified in the sunflower. Fragment duplication is the main cause of the expansion of PYL genes in the sunflower genome. Based on phylogenetic analysis, HaPYL genes were divided into three subfamilies. Members in the same subfamily share similar protein motifs and gene exon-intron structures, except for the second subfamily. Tissue expression patterns suggested that HaPYLs serve different functions when responding to developmental and environmental signals in the sunflower. Exogenous ABA treatment showed that most HaPYLs respond to an increase in the ABA level. Among these HaPYLs, HaPYL2a, HaPYL4d, HaPYL4g, HaPYL8a, HaPYL8b, HaPYL8c, HaPYL9b, and HaPYL9c were up-regulated with PEG6000 treatment and NaCl treatment. This indicates that they may play a role in resisting drought and salt stress in the sunflower by mediating ABA signaling. Our findings provide some clues to further explore the functions of PYL genes in the sunflower, especially with regards to drought and salt stress resistance.

Introduction

Climate change has significantly affected global agricultural production and has raised concerns among biologists about food security (Raza et al., 2019; Zhou et al., 2023). Environmental stresses are the main constraints limiting plant growth, production, and distribution (Osakabe et al., 2012; Osakabe, Osakabe & Shinozaki, 2013). Specifically, drought and soil salinity are the most devastating abiotic stresses affecting agriculture today (Zhang et al., 2006). Drought stress is the most common environmental factor limiting crop productivity and global climate change is increasing the frequency of severe droughts (Dai, 2013). In addition, salinity is predicted to affect 7% of terrestrial land (Hopmans et al., 2021). The area of land used for food production is predicted to decrease as a result of the dramatic loss and degradation of arable land due to increased salinity and drought. To maintain high biological yields of crops and to meet the growing demand for food, the breeding of climate-resilient crops that are resistant to extreme environments (such as drought, salinity, and high temperatures) is urgently needed (Alexandratos & Bruinsma, 2012; Dhankher & Foyer, 2018; Misra, 2014).

The sunflower is the fifth largest oilseed crop in the world. Globally, over 27.36 million hectares are planted, producing a world average yield of 204 tons per unit area (Meena & Sujatha, 2022). With an oil content of 40–50% and a protein content of 17–20%, the sunflower has considerable potential to bridge the gap between global production and the demand for edible oils and animal feed (Hussain et al., 2018). The sunflower is mainly grown in arid, semi-arid, or saline areas and is an attractive stress-resistant crop that grows in harsh environments (Hladni et al., 2022; Kane & Rieseberg, 2007). In the future, sunflower cultivation is predicted to expand into marginal areas with low soil fertility and unfavorable climates (Seiler, Jan & Hu, 2010). Several reports showed that drought and salinity can reduce seed yield, oil production, and oil quality in sunflowers (Akram, Ashraf & Al-Qurainy, 2011; Di Caterina et al., 2007; Ibrahim, Faisal & Shehata, 2016; Noreen et al., 2019; Oraki & Aghaalikhana, 2012). Therefore, breeding for drought and salinity-tolerant cultivars will contribute to the expansion of the sunflower into marginal areas and consequently increase the stability of the global supply of sunflowers. The availability of sunflower regeneration systems and gene editing techniques make it possible to breed drought and salt-tolerant sunflower cultivars (Darqui et al., 2021; Kumar et al., 2023; Li et al., 2021; Sheri, Muddanuru & Mulpuri, 2021). However, this requires finding genes that confer salt and drought tolerance to the plant.

As the main types of environmental stresses, drought and salt stress lead to varying degrees of dehydration in plants, which induce a hyperosmotic signal and trigger abscisic acid (ABA) biosynthesis (Yoshida, Mogami & Yamaguchi-Shinozaki, 2014; Zhu, 2002). ABA signaling is an essential process that regulates plant tolerance to drought and salt stress (Ali, Pardo & Yun, 2020; Lim et al., 2012; Vishwakarma et al., 2017). The PYR1/PYL/RCAR (PYRABACTIN RESISTANCE1/PYR1-LIKE /REGULATORY COMPONENTS OF ABA RECEPTORS) receptors are core components of ABA signal transduction and play an important role in drought and salt stress tolerance by mediating ABA signaling (Fidler et al., 2022; Rehman et al., 2021; Sun, Fan & Mu, 2017). Genetic evidence showed that the overexpression of AtPYLs, including PYL4, PYL5, PYL7, PYL8, and PYL9, increased drought tolerance in Arabidopsis thaliana (Lee et al., 2013; Pizzio et al., 2013; Santiago et al., 2009; Zhao et al., 2016). Ectopic expression of OsPYL5 enhanced drought and salt tolerance in rice (Kim et al., 2014). FYVE1 decreased salt tolerance in A. thaliana, which was associated with the degradation of the PYR1 and PYL4 receptors (Pan et al., 2020). The overexpression of ABA receptors in wheat increased sensitivity to ABA, decreased the transpiration rate, and increased the photosynthetic rate, thereby significantly reducing the lifetime water consumption of wheat (Mega et al., 2019). These results confirmed the crucial role of PYL genes in drought and salt stress resistance. The study of PYL genes’ responses to abiotic stresses is important in analyzing their role in plant tolerance to abiotic stress via the ABA signaling pathway. However, the PYL gene family has not been studied in the sunflower, which hinders the analysis of sunflower development and abiotic stress resistance.

In this study, we comprehensively described the sunflower’s PYL gene family in terms of its member identification, phylogeny, gene structure, protein properties, protein motifs, cis-regulatory elements, and tissue expression patterns. The response of HaPYL genes to drought and salt stress was explored using PEG6000, NaCl, and ABA treatment. In this way, several critical PYL genes responsive to drought and salt stress were selected. The results of this study provided valuable information for further studies on the role of HaPYL genes in drought stress and salt stress tolerance. This is needed to establish further insights into the biological functions HaPYLs serve in abiotic stress resistance in the sunflower and their application for environmental adaptation.

Materials & Methods

Genome-wide identification of HaPYL gene family

Genome-wide protein sequences of the sunflower were downloaded from the Ensembl Plants database (https://plants.ensembl.org/index.html). The HMM model file (PF10604) of the polyketid_cyc2 structural domain was downloaded from the Pfam database (http://pfam.xfam.org/) and used to screen PYL proteins from all protein sequences in the sunflower using HMMER 3.0 software (default parameter settings) (Finn, Clements & Eddy, 2011). The protein sequences of AtPYLs were used for the identification of sunflower PYLs by using BLASTP (Altschul et al., 1990) and the E-value threshold of the BLAST program was set at less than e−5. These results were summarized and preliminary sequences of PYL proteins in the sunflower were thus derived. These sequences were then submitted to SMART (http://smart.embl-heidelberg.de/), CDD (https://www.ncbi.nlm.nih.gov/cdd), and PFAM (http://pfam.xfam.org/) to confirm their family identity. The names of HaPYLs were based on the NCBI protein annotations and HaPYL proteins with the same annotations were differentiated by lowercase letters according to their chromosomal location. Molecular weight (MW), theoretical isoelectric point (pI), and protein length (aa) were manually calculated using the ExPASy server (http://web.expasy.org/). The subcellular localization prediction analysis was conducted using the PLoc server (http://www.csbio.sjtu.edu.cn/bioinf/Cell-PLoc-2/).

Phylogenetic analysis

The phylogenetic tree was built using MEGA 11 software (Altschul et al., 1990), by alignment of the protein sequences of PYLs in A. thaliana, Nicotiana tabacum, Zea mays, Oryza sativa, and Helianthus annuus (File S2). The multiple alignment of these sequences was performed using ClustalW (Thompson, Gibson & Higgins, 2003), and an unrooted phylogenetic tree was constructed using the neighbor-joining method with 1,000 bootstrap replicates.

Chromosome location and collinearity analysis

The genome annotation files of the sunflower were downloaded from the Ensembl Plants database (https://plants.ensembl.org/index.html). The chromosome position of HaPYL genes and the chromosome length were obtained and visualized using MapChart software (Voorrips, 2002). Homology was detected with Mcscanx using the default parameter settings (Wang et al., 2012). The sunflower protein-coding genes and the whole genome were compared to calculate all possible chromosomal co-linkage blocks. TBtools was used to highlight identified PYL co-linkage gene pairs (Chen et al., 2020).

Analysis of conserved motif, gene structure, and protein properties

The gff3 annotation files for the sunflower and A. thaliana PYL genes were obtained and the exon-intron structure of the PYL genes were then analyzed using GSDS 2.0 (http://gsds.gao-lab.org/). The conserved motifs of the PYL proteins were predicted using MEME (http://alternate.meme-suite.org/tools/meme), with the maximum number of motifs set to eight. Motifs and exon-intron structures were placed adjacent to their respective PYLs according to the subfamily characteristics of their phylogenetic relationships, which were also visualized using TBtools (Chen et al., 2020).

The multiple sequence alignment of the HaPYL proteins was performed using ClustalW. The crystal structure of A. thaliana, PYR1, (Protein DataBank Code 3K90) was used as a model and the alignment results were treated using Espript (https://espript.ibcp.fr/ESPript/cgi-bin/ESPript.cgi), to predict the secondary structure of the HaPYL proteins.

Promoter analysis, protein function annotation, and protein interaction network analysis

The corresponding 2,000-bp sequences located above the transcription start site of the HaPYLs were obtained from the sunflower genome sequence file (File S3). The cis- regulatory elements in these sequences were predicted using the PlantCARE web server (http://bioinformatics.psb.ugent.be/webtools/plantcare/html/).

The protein sequences of the HaPYLs were compared to items in the NCBI nonredundant (NR) protein database (Pruitt et al., 2012), Swiss-Prot (Apweiler et al., 2004), the Protein family (Pfam) database (Finn et al., 2014), GO (Finn et al., 2014), the clusters of orthologous groups (COG) database (Tatusov et al., 2000), the eukaryotic orthologous groups (KOG) database (Koonin et al., 2004), and the Kyoto Encyclopedia of Genes and Genomes (KEGG) database (Kanehisa et al., 2004). The GO enrichment analysis was performed on the annotated HaPYL proteins.

The STRING database (http://string-db.org/cgi) was used to search and analyze the interactions between HaPYL proteins and other sunflower proteins, with a set minimum required interaction score of 0.7 and a maximum number of interactors set at 20. The interactions were displayed through Cytoscape software (Doncheva et al., 2018).

Tissue expression profile of PYL genes in sunflower

We filtered the HaPYLs expression data from the full-growth period tissue expression data of the sunflowers that had been processed in our lab. Expression levels of the HaPYLs were quantified based on their fragments per kilobase of exon per million reads mapped (FPKM) values in Cufflinks using the default parameters (Ghosh & Chan, 2016). The results were presented in a heatmap obtained through the pheatmap package in the R platform.

Plant materials, abiotic stress and ABA treatments

All sunflower seedlings were grown under a 16 h light/8 h dark photoperiod at 28 °C. Sunflower seedlings were germinated and grown in soil under normal watering conditions until the three-leaf stage. Plants were then soaked in water containing 10% PEG6000 to simulate drought or 100 mM NaCl to simulate salinity. In addition, the roots and leaves of the seedlings were treated with 100 µM of ABA solution for ABA treatment. Leaf samples were collected at 3, 6, 12, 24, and 48 h after PEG6000 and ABA treatment and at 3, 6, 12, 24, and 75 h after NaCl treatment. All collections were immediately frozen in liquid nitrogen and stored in a −80 °C freezer for RNA isolation. Each treatment sample consisted of at least nine biological replicates.

RNA Extraction and quantitative real-time RT-PCR

Total RNA was extracted from each sample using OMEGA E.Z.N.A.® Plant RNA Kit. The RNA concentration was measured using Nanodrop2000 and the quality of RNA was evaluated by electrophoresis. The corresponding cDNA was synthesized from the total RNA using the Takara PrimeScript™ RT reagent Kit with gDNA Eraser. Quantitative primers for HaPYLs were designed in Primer Premier 5.0 and are listed in File S5. The eF1 gene in sunflowers was considered an internal reference gene. The qRT-PCR was performed using SYBR Green Supermix (Bio-Rad, Hercules, CA, USA) on the CFX96 Real-time System (Bio-Rad). Each experiment was conducted at least three times. The relative expression levels of these genes were calculated on the basis of the 2−ΔΔCt method (Livak & Schmittgen, 2001) and were normalized against the eF1 gene.

Statistical analysis

The data obtained by qRT-PCR was the mean ± SE of three individual experiments with three replications. Histograms of the HaPYL gene expression were plotted using GraphPad Prism (GraphPad Prism for Windows version 9.0; GraphPad Software, San Diego, CA, USA), and the student’s t-test was performed to compare the variations of the HaPYL expression value at each indicated time and 0 h based on significance levels of P < 0.05, P <  0.01, and P < 0.001 (De Winter, 2019).

Results

Genome-wide identification of HaPYLs in sunflower

A total of 19 HaPYL members were identified in the sunflower genome. The lengths of HaPYL protein sequences vary from 172 (HaPYL9b) to 219 (HaPYR1a) amino acids. The molecular weight (MW) varies from 19.09 to 24.27 kDa, and the Isoelectric Points (pI) vary from 5.44 to 8.72, with an average of 6.24. All HaPYL proteins are hydrophilic, yet most are unstable and susceptible to degradation. Subcellular localization prediction showed that most HaPYL proteins are located in the cytoplasm and a few are found in the nucleus and chloroplast (Table 1).

Table 1 Genomic information and protein characterization of sunflower HaPYL gene family.

Gene_name	Gene_ID	Chromosomal position	Amino acid/aa	Isoelectric point [PI]	Molecular weight/kDa	Instability index	Aliphatic index	Grand average of hydropathicity (GRAVY)	Subcellular localization	
		Number	Start	End								
HaPYR1a	HanXRQr2_Chr06g0242911	6	7,648,694	7,649,349	219	5.44	24.27	38.81	83.56	−0.308	Nucleus	
HaPYR1b	HanXRQr2_Chr12g0530941	12	18,226,090	18,227,034	199	5.74	22.32	28.82	87.69	−0.309	Nucleus	
HaPYL2a	HanXRQr2_Chr03g0121241	3	143,704,924	143,705,484	186	5.35	20.81	30.5	96.83	−0.135	Cytoplasm	
HaPYL2b	HanXRQr2_Chr07g0285411	7	28,728,922	28,729,686	193	5.5	21.61	36.18	87.2	−0.29	Cytoplasm	
HaPYL2c	HanXRQr2_Chr13g0596431	13	118,972,860	118,973,797	186	5.66	20.79	44.47	90.48	−0.209	Cytoplasm	
HaPYL2d	HanXRQr2_Chr16g0742371	16	71,822,459	71,822,989	186	6.04	20.56	28.85	88.39	−0.218	Cytoplasm	
HaPYL4a	HanXRQr2_Chr02g0070891	2	127,794,460	127,796,663	176	6.53	19.09	24.83	91.76	−0.097	Cytoplasm	
HaPYL4b	HanXRQr2_Chr05g0211711	5	96,693,894	96,694,499	201	6.91	22.56	50.6	88.46	−0.205	Cytoplasm	
HaPYL4c	HanXRQr2_Chr07g0313141	7	144,016,945	144,017,941	209	6.88	23.20	50.03	90.67	−0.144	Cytoplasm	
HaPYL4d	HanXRQr2_Chr08g0334961	8	41,057,454	41,058,282	204	6.29	22.62	41.65	94.41	−0.169	Cytoplasm	
HaPYL4e	HanXRQr2_Chr08g0342731	8	675,847	676,829	206	6.28	22.40	41.9	83.59	−0.227	Cytoplasm	
HaPYL4f	HanXRQr2_Chr10g0429361	10	40,056,822	40,057,968	205	6.02	22.25	48.37	93.07	−0.141	Cytoplasm	
HaPYL4g	HanXRQr2_Chr14g0666021	14	166,363,082	166,364,056	206	6.52	22.59	47.61	86.84	−0.197	Cytoplasm	
HaPYL8a	HanXRQr2_Chr06g0241551	6	5,026,265	5,028,331	189	5.46	21.08	44.91	94.76	−0.331	Nucleus	
HaPYL8b	HanXRQr2_Chr07g0316871	7	149,070,093	149,072,027	191	6.06	21.69	51.01	94.82	−0.405	Nucleus	
HaPYL8c	HanXRQr2_Chr17g0780891	17	3,516,336	3,520,004	189	6.19	21.20	48.35	93.7	−0.275	Chloroplast	
HaPYL9a	HanXRQr2_Chr04g0182281	4	188,532,978	188,533,979	184	7.17	20.60	55.73	90.43	−0.215	Chloroplast	
HaPYL9b	HanXRQr2_Chr06g0246931	6	15,659,941	15,661,299	172	8.72	19.31	46.45	92.27	−0.3	Chloroplast	
HaPYL9c	HanXRQr2_Chr15g0671331	15	675,945	676,565	193	5.82	21.42	46.08	96.27	−0.213	Chloroplast	

Phylogenetic analysis of PYL gene family in Arabidopsis, maize, tobacco, rice and sunflower

An unrooted neighbor-joining (NJ) tree was constructed using MEGA software with 29 NtPYLs from tobacco, 14 AtPYLs from Arabidopsis, 13 OsPYLs from rice, 13 ZmPYLs from maize, and 19 HaPYLs from sunflowers (Fig. 1). In Arabidopsis, the PYL gene family could be classified into three subfamilies: Subfamily I, Subfamily II, and Subfamily III (Ma et al., 2009; Park et al., 2009). The phylogenetic analysis revealed that 19 HaPYLs could be grouped with their orthologous PYLs from Arabidopsis. According to the established phylogenetic tree, HaPYL8 clade and HaPYL9 clade belong to Subfamily I. HaPYL4 clade is contained in Subfamily II. HaPYL2 clade and HaPYR1 clade are included in Subfamily III (Fig. 1). The three subfamilies have roughly the same number of HaPYL members.

Figure 1 Phylogenetic analysis of PYL proteins in Arabidopsis, rice, sunflower, tobacco, and maize.

PYL proteins from sunflower (19), Arabidopsis (14), rice (13), maize (13), and tobacco (29) were used to constructed an unrooted neighbor-joining trees with 1,000 bootstrap repetitions. The tree was classified into three subfamilies: Subfamily I, Subfamily II, and Subfamily III, indicated in red, green, and blue, respectively. Different species are marked with different colored symbols; yellow pentagrams for HaPYLs, blue boxes for OsPYLs, red right-facing triangles for NtPYLs, purple left-facing triangles for ZmPYLs, and green circles for AtPYLs. HaPYLs are bolded for clarity.

Chromosomal locations and intraspecific collinearity of HaPYLs

HaPYL genes are unevenly distributed on 15 of 17 chromosomes in the sunflower. The chromosomes containing the most HaPYL genes are chr. 6 and chr. 7, with a total number of three genes. This was followed by chr. 8 with a total of two. No HaPYL genes were found on chr. 1 and chr. 9 (Fig. 2).

Figure 2 Distribution of HaPYL genes on sunflower chromosomes.

Chromosome numbers are shown at the top of each chromosome. The strip on the chromosome indicates the location of the HaPYL gene. The scale ruler on the left shows the length of the chromosome and the approximate location of the HaPYL gene on the chromosome in mega base (Mb).

Most HaPYLs were located on the corresponding homologous chromosomes. Fragment duplication is the main reason for the expansion of HaPYL genes. Gene duplications occurred within the same subfamily (e.g., HaPYL2c-HaPYL2a, HaPYL2a-HaPYL2b, HaPYL2c-HaPYL2b, HaPYL4b-HaPYL4d, HaPYL4g-HaPYL4c, HaPYL4g-HaPYL4d, HaPYL8c-HaPYL8a, and HaPYL9a-HaPYL9b), but also between subfamilies (e.g., HaPYL4f-HaPYL9c, HaPYL8c-HaPYL4a, HaPYL9c-HaPYL2d, and HaPYL4f-HaPYL2d) (Fig. 3).

Figure 3 Circle graph showing collinearity of the HaPYL gene family.

The sunflower chromosomes form a circle with chromosomal names shown on the periphery of the ring. The PYL genes are distributed on the inner side of the ring based on their chromosomal locations, with the blue lines highlighting gene covariance.

Gene structure and conserved motifs of HaPYLs

The exon-intron structure is an important feature of gene evolution and can provide some insights into the gene’s functional diversification. Based on the exon-intron structure, HaPYL genes can be divided into intronless and intronic clusters (Fig. 4B). Most HaPYL genes belong to the intronless clade, while the intron clade is dominated by the HaPYLsubfamily I (Fig. 4B). HaPYL genes have similar exon-intron structure with their homologous AtPYL genes in each subfamily, except for HaPYR1a and HaPYL2b of Subfamily III and HaPYL4a of Subfamily II. These results confirmed their close evolutionary relationship and the classification of the subfamilies.

Figure 4 Phylogenetic tree, gene structure, and conserved motifs of PYLs in sunflower and Arabidopsis.

(A) Phylogenetic tree of PYLs in sunflower (HaPYLs) and Arabidopsis (AtPYLs). The tree was constructed using the neighbor-joining method. Red, green, and blue boxes mark Subfamily I, II, and III, respectively. (B) Exon/intron structure of PYL genes. Blue trips indicate untranslated regions (UTRs), orange trips indicate protein-coding domains (CDSs), and lines indicate introns. (C) Distribution of conserved motifs of PYL proteins. Motifs are characterized by different colored boxes and the corresponding protein sequences are depicted in (D). The motif position and CDS length of each PYL genes can be calculated according to the scale at the bottom.

As the structural domain is the basic unit needed for a protein to perform its function, the conserved motifs in the proteins encoded by PYL genes were subsequently analyzed (Fig. 4C). The results showed that eight motifs were identified, of which motifs 1, 2, and 3 are present in all HaPYL proteins. This constitutes a star-related lipid transfer (START) domain (Figs. 4C and 4D). Motif 4 and motif 6 were unique to Subfamily III and I, respectively. HaPYL proteins in subfamily II have diverse motifs, with HaPYL4e, HaPYL4a, and HaPYL4f having motif 8; HaPYL4b, HaPYL4c, HaPYL4d, and HaPYL4g having motif 7; HaPYL4c, HaPYL4b, HaPYL4d, HaPYL4f, and HaPYL4g having motif 5 along with HaPYL proteins of Subfamily I. HaPYL proteins grouped in the same subfamily exhibited similar motif features, indicating that they may have similar functions. The presence of subfamily specific motifs may represent subfamily-specialized biological functions.

Conservative sequences and CL2 regions/loops of HaPYLs

Amino acid alignment analysis showed that all identified HaPYL proteins have a similar helical-grip organization with three α-helices separated by seven β-sheets and several conserved CL regions/loops (Fig. 5). Previous studies showed that the CL2 region/loop of PYL proteins, especially the No. 3 and No. 4 amino acid residues in the CL2 region, was critical for the monomeric or dimeric state of PYL-PP2C interactions, ABA dependence, and activity of PYLs (Hao et al., 2011; Park et al., 2009; Santiago et al., 2012). Amino acid residues of conserved CL2 regions in the HaPYLs showed a certain degree of similarity and polymorphism. In the sunflower, the combinations of the No. 3 and 4 residues in the CL2 region were VK and VR, VI and VV, and VI and VM in the HaPYL Subfamily I, II, and III, respectively (Fig. 5). In addition to the CL2 region, the other crucial amino acid residues of HaPYL proteins that bind to ABA, including K59, A89, E94, R116, Y120, S122, and E141, are conserved.

Figure 5 HaPYL protein multiple sequence alignment and secondary structure.

The 19 HaPYL protein sequences were aligned using ClustalW and their secondary structures were predicted using the Espript online service based on the Arabidopsis PYR1 (Protein DataBank Code 3K90) crystal structure. The secondary structure elements are placed above the primary sequence. Black helices and arrows represent α-helix and β-folded sheets, respectively, and T indicates a corner turn. Black boxes mark the important CL2-gate and CL3-latch structural domains of the PYL protein that bind and lock to ABA. Black asterisks denote key and conserved amino acid residues in the PYL protein sequence involved in ABA binding.

Gene Ontology enrichment and protein interaction networks of HaPYLs

To determine the function HaPYLs exert, GO annotation and GO enrichment analyses were performed using three categories of GO terms: biological process (BP), molecular function (MF), and cellular component (CC). The available GO enrichment results showed that HaPYLs in the CC categories are enriched in the nucleus (GO:0005634), cell membrane (0016020), cell membrane constituents (GO:0016021), and cytoplasm (GO:0005737). The BP category is mainly associated with abscisic acid signaling pathway activation (GO:0009738) and protein serine/threonine phosphorylation activation regulation (GO:0080163). Protein phosphorylation inhibitor activation (GO:0004864), receptor activation (GO:004872), negative regulation of catalytic activity (GO:0043086), and binding of abscisic acid (GO:0010427) are the most enriched functions in the MF category (File S6).

To predict the potential targets of HaPYL proteins, the amino acid sequences of HaPYLs were used as input for STRING against all sunflower proteins available in the protein interactions database, with a set maximum number of interacting proteins of 20 and a minimum interaction score of 0.7. As expected, most proteins interacting with HaPYLs, such as PP2C and SnRK2, are important components of ABA signaling complexes (Fig. 6). Based on functional annotations, the proteins interacting with HaPYLs can be divided into four categories, including 18 phosphatase 2C family proteins (four PP2C24, four PP2C37, two PP2C75, HAI2, PP2C38, PP2C51, PP2C6, ABI2, HAB1, PP2C16 and a putative protein-serine/threonine phosphatase), 16 serine/threonine protein kinases (five SRK2A, four SRK2E, SRK2I, SAPK2, three SAPK3, SOS2, and a protein kinase domain-containing protein), an abscisic acid 8′-hydroxylase CYP707A2, and MPK4 (Files S7, S8). CYP707A2 specifically interacted with the HaPYL2a protein. HaPYLs from the same clade have the same interacting proteins. The HaPYL2 clade proteins have the highest number of interacting proteins, with 18, followed by those of Subfamily III and HaPYL1a, with 17. The HaPYL4 clade proteins have the lowest number of interacting proteins, with 11 (Fig. 6). Interacting proteins of HaPYL1b were not detected using the screening requirement of the interaction score of 0.7, so HaPYL1b was screened with a lower interaction score of 0.6. This resulted in finding four interacting proteins.

Figure 6 HaPYL’s protein interaction network.

STRING online service was utilized to search for proteins in the sunflower that can interact with HaPYL proteins and the maximum number of interacting proteins was set to 20 with an interaction score greater than 0.7. However, HaPYL1b was screened with a lower interaction score of 0.6. The graph was drawn by Cytoscape software. The pink, blue, and orange colors denote PYLs, PP2Cs, and SnRKs, respectively. The connecting lines indicate the interactions.

Expression pattern of HaPYLs in tissues at different developmental stages

To further understand the potential functions of HaPYL genes in sunflower development, their expression levels in tissues at different developmental stages were analyzed (Fig. 7). More than one HaPYL gene was expressed in sunflower tissues at any developmental stage, implying that multiple PYL genes are required for coordinated function during sunflower development. HaPYL8a, HaPYL8b, HaPYL8c, HaPYL9b, HaPYL9c, HaPYL4c, HaPYL4f, and HaPYR1a were expressed at higher levels in almost all sunflower tissues, indicating their important role in regulating sunflower biological processes. In contrast, the transcript abundance of HaPYL4g, HaPYL4e, HaPYL2d, and HaPYL2b was very low in all organs (Fig. 7), indicating that they may serve a limited role in sunflower development. HaPYR1a, HaPYL4c, HaPYL4d, HaPYL4e, and HaPYL4g were highly expressed in young roots and HaPYL9a and HaPYL9b were highly expressed in leaves, where they may play an important role in regulating the development of nutrient organs. HaPYL8a was consistently highly expressed in st2-st4 (early stages of flower development), indicating their critical roles in flower initiation and polarization. HaPYL8b and HaPYL8c had higher expression levels at seed development than other HaPYL genes (st7-st8), suggesting that they may promote the development and maturation of sunflower seeds.

Figure 7 Expression pattern of HaPYL genes in sunflower tissues at different developmental stages.

The relative transcript abundance of HaPYL genes were examined by microarray of sunflower young roots, young stems, young leaves, roots, stems, leaves, and flowers at different stages of development (st2-st8). st2 was the emergence of the floret primordium, st3 was the initiation of the floral organ, st4 and st5 were the development of the floral organ, st6 was the flowering stage, st7 was the embryo nurturing stage, and st8 was the maturation of the seed. The results were visualized in a heat map. Clusters were arranged according to gene expressions. The color scale is represented after homogenization with FPKM values and the mapping is shown in the legend on the right side of the figure.

Cis-regulatory elements in the promoter of HaPYLs

To explore the response of HaPYL genes to developmental and environmental signals, the promoter cis-regulatory elements of the HaPYL genes were analyzed. The results showed that four abiotic stress response elements were detected in the 2,000 bp sequences upstream of the HaPYL gene promoters, including the defense and stress response, low-temperature response, anaerobic induction, and drought induction (Fig. 8). In addition, a variety of phytohormone responses were found in the HaPYL gene promoter sequences, including abscisic acid, auxin, methyl jasmonate (Me JA), gibberellin, and salicylic acid. Abscisic acid response occurred in all HaPYL gene promoter sequences (Fig. 8).

Figure 8 Cis-regulatory elements in the HaPYL gene promoter.

Different colored squares indicate distinct response elements. See the legend on the right for mapping.

Response of HaPYLs to ABA, drought stress and salinity stress

To confirm the role of HaPYLs in ABA reception and to understand their response to increased ABA content in the plant, transcriptional changes of all HaPYLs were measured by exogenous ABA application. The transcript levels of most HaPYL genes were up-regulated after ABA treatment, while HaPYL4a and HaPYL2b were suppressed (Fig. 9). The transcript levels of HaPYL4c, HaPYL4d, and HaPYL4f were suppressed by ABA at first, but were later up-regulated. These results indicated that most HaPYL genes can respond to increased ABA levels but they might have different responses to abiotic stress. HaPYL8b, HaPYL8c, HaPYL4e-HaPYL4g, and HaPYL2a were the first responders to ABA treatment and transcript abundance increased after 3 h of ABA treatment (HaPYL4f was down-regulated), suggesting that they were sensitive genes for sensing ABA signals.

Figure 9 Response of HaPYL genes to PEG6000, NaCl, and ABA treatment.

The PEG6000, ABA, and NaCl concentration used for treatments was 10%, 100 µM, and 100 mM. HaeF1 was used as a reference to calculate HaPYLs relative expression values by applying the 2−ΔΔCt method. The values represent the mean ± SE of the three biological replicates. Student’s t-test was conducted to analyze the difference of the expression values between each HaPYL at each set time and 0 h, with the difference in significance indicated by the asterisk * (∗p < 0.05, ∗∗p < 0.01, ∗∗∗p < 0.001).

To explore the response of HaPYL genes to drought and salt stress, the expression levels of HaPYLs were analyzed in sunflower seedlings at specific times after 10% PEG6000 and 100 mM salt treatments (Fig. 9). The transcript levels of HaPYL2a, HaPYL4d, HaPYL4g, HaPYL8a, HaPYL8b, HaPYL8c, HaPYL9b, and HaPYL9c were up-regulated in both PEG6000 and NaCl treatments while HaPYL4a, HaPYL4e, and HaPYL4f were down-regulated in both treatments. HaPYL4b was not detected in either of the two treatments. HaPYR1a, HaPYR1b, HaPYL2c, HaPYL2d, HaPYL4c, and HaPYL9c showed different response patterns between the two treatments and their responses to the NaCl treatment were more complex (Fig. 9). HaPYL2b and HaPYL9a presented opposite changes in expression levels in PEG6000 and NaCl treatment. Among the HaPYLs with higher expression levels under PEG6000 and NaCl treatment, the earliest responses began 3 h after treatment, including HaPYL2a and HaPYL8b with PEG6000 treatment and HaPYL8a, HaPYL8b, and HaPYL9b with NaCl treatment (Fig. 9). These HaPYL genes could be the early response genes for drought and salt stress. The expressions of HaPYL4g and HaPYL8b were up-regulated at the late stage of PEG6000 treatment (48 h). The expressions of HaPYL8a, HaPYL8b, HaPYL8c, HaPYL9c, HaPYL4d, and HaPYL4g were up-regulated at the late stage of NaCl treatment (75h). These could be important for the adaptive regulation of the sunflower in extreme drought and salt stress conditions.

Discussion

Abscisic acid is an important hormone for plant growth and development, and it could be a response to many environmental stresses (Kishor et al., 2022; Kumar et al., 2019), through the ABA signaling pathway. The PYL receptor is a core component of ABA signaling and essential for activating the ABA signaling cascade response. Considering the importance of ABA receptors (PYLs) in the ABA signaling pathway, studies of the PYL genes have been reported in several species, including A. thaliana (Ma et al., 2009; Park et al., 2009), rice (Yadav et al., 2020), sweet cherry (Zhou et al., 2023), poplar (Yu et al., 2016), and rubber tree (Guo et al., 2017). However, work on PYL genome-wide characterization in the sunflower is limited, which hampers the study of their developmental functions and roles in abiotic stress resistance. In this study, we used bioinformatics and experimental approaches to characterize the sunflower PYL gene family and depict their potential functions in sunflower development in response to drought and salt stress.

In this study, 19 HaPYL genes were identified in the sunflower genome. Compared with the number of PYL genes in other reported diploid plants, including A. thaliana (14) (Ma et al., 2009; Park et al., 2009), rice (13) (Yadav et al., 2020), sweet cherry (11) (Zhou et al., 2023), poplar (14) (Yu et al., 2016), and rubber tree (14) (Guo et al., 2017), the number of HaPYLs in the sunflower is greater. This implies that HaPYLs may be important in sunflower development and used in environmental adaptation. ExPaSy analysis, aimed at characterizing HaPYL proteins, showed that all HaPYL proteins are hydrophilic. There is no significant difference in sequence length and physical properties of HaPYL proteins, yet they do differ in stability and most are easily degraded, indicating that the functional differentiation of HaPYL proteins may be less related to their physical properties than to their chemical properties. Subcellular localization predictions showed that most HaPYL proteins are located in the cytoplasm and few are found in the chloroplast and nucleus.

According to phylogenetic analysis, HaPYL proteins can be categorized into three subfamilies, with HaPYL8 clade and HaPYL9 clade in Subfamily I, HaPYL4 clade in Subfamily II, and HaPYL2 clade and HaPYR1 clade in Subfamily III (Fig. 1). The number of members is essentially equal in the three subfamilies. The HaPYL4 clade has the highest number of clade members followed by the HaPYL2 clade, which may be related to their status as the oldest clade of the PYL gene family (Yang et al., 2020). The protein sequence similarity between sunflower HaPYLs and tobacco NtPYLs is higher than between HaPYLs and other PYLs, which may be related to the fact that sunflower and tobacco plants are both semi-drought tolerant. The Mcscanx analysis revealed that fragment duplication is the main driving force for the expansion of PYL genes in the sunflower genome.

Gene structure and protein motif analysis provided further clues into the evolution and functional divergence of HaPYL genes. HaPYL genes of the same subfamily have similar gene structures and protein motifs, confirming their close evolutionary relationship and thus the classification of the subfamilies (Fig. 4). Based on the presence or absence of introns, HaPYL genes can be divided into intronic and intronless clades. Subfamily I (HaPYL4a, HaPYL2b, and HaPYR1a) belongs to the intronic clade and the remaining HaPYLs belong to the intronless clade (Fig. 4B), which indicates a different evolutionary history of the two clades. Compared with the intronless clade genes, the functional regulation of the intronic clade genes is more flexible because introns play a vital role in post-transcriptional regulation of gene expression through splicing-dependent and splicing-independent intron-mediated enhancement of mRNA accumulation (Gallegos & Rose, 2019). HaPYL genes with introns may be involved in a wide range of biological processes in the sunflower.

Motif 1, motif 2, and motif 3 are present in all HaPYL proteins, which forms a star-related lipid-transfer (START) domain (Fig. 4C). The START domain is characterized by the presence of a central β-sheet surrounded by N- and C-terminal α-helices, with long C-terminal α-helices stacked tightly on the β-sheet. The helix grip folds over to create a large cavity that serves as the ABA binding pouch (Figs. 4C, 4D and 5) (Hao et al., 2011; Yin et al., 2009). These motifs are conserved during HaPYL protein evolution and are essential for their function. Motif 6, motif 7/8, and motif 4 are unique to PYL subfamilies I, II, and III, respectively. Motif 5 is specific to subfamily I and most genes of Subfamily II. These findings showed that HaPYL proteins have distinct subfamily characteristics. Motif 4, motif 5, and motif 6 are located near the star-associated lipid transfer (START) structural domain. Motif 7 and motif 8 are located at the N-terminus of the PYL protein. Motif 4, motif 5, and motif 6 may affect the binding ability of the PYL protein to ABA because of their position, while motif 7 and motif 8 may not. In addition, these subfamily-characterized motifs have no relationship to their subcellular localization, so these motifs do not function as subcellularly localized signal peptides. Although the roles of these subfamily-characterized motifs are not yet known to us, they are helpful for the study of the evolution of the PYL gene family. The motifs of Subfamily II have the most diverse composition among the three subfamilies. Subfamily II may provide more clues to the origin and evolution of the PYL gene family in the sunflower.

ABA signaling is inhibited by PP2Cs, and PYLs stop this inhibition in an ABA-dependent manner. Most PYLs exist as homodimers and each PYL protomer contains a ligand-binding pocket formed by four conserved loops (CL1-CL4). Upon binding to (+)-ABA molecules, the conserved ring CL2 undergoes a conformational change to provide a new binding surface and the ABA-PYL complex forms a 1:1 heterodimer with PP2C through the newly formed binding surface. The CL2 region is located above the active site of PP2C, preventing substrates from accessing PP2Cs and thus mitigating PP2Cs-mediated inhibition of SnRK2. The CL2 region is a vital functional region of PYL, and is essential for PYL-mediated ABA signaling (Hao et al., 2010; Yin et al., 2009; Zhang et al., 2015).

Multiple sequence alignment revealed a certain degree of similarity and polymorphism of amino acid residues in the CL2 region of HaPYL proteins. In A. thaliana, soybean, and tobacco, the combinations of the No. 3 and No. 4 amino acid residues in the CL2 region were conserved. They were VI and VV, VK and LK, VV and LV in PYL Subfamily I, II, and III, respectively (Bai et al., 2019; Bai et al., 2013; Santiago et al., 2012). In the sunflower, the combinations were VK and VR, VI and VV, and VI and VM in HaPYL Subfamily I, II, and III, respectively (Fig. 5). The different PYL-ABA-PP2C binding sites in HaPYL proteins imply HaPYL proteins may bind to ABA and PP2C in a more flexible and diverse way than previously thought. HaPYL proteins provide new models for structural studies of ABA-PYL-PP2C binding.

To predict the potential binding targets of HaPYL proteins, all sunflower proteins in the database that can interact with HaPYL proteins were investigated using the STRING online web service, setting the maximum number of interacting proteins at 20 and the minimum interaction score at 0.7. The results showed that almost all the proteins that could interact with HaPYL proteins were from the PP2C gene family. Proteins that could interact with PP2C proteins besides HaPYL proteins were mainly SnRK2 proteins (Fig. 6), which indicated that the way HaPYL proteins function is preserved in the sunflower and is primarily involved in the process of ABA signaling. The results of the gene ontology annotation and enrichment of HaPYL genes also illustrated this. HaPYL proteins of the same clade have the same interworking proteins, suggesting that HaPYL proteins of the same clade may have evolved with conserved interacting structures and binding sites. This is consistent with the results of their secondary structure analysis (Figs. 5 and 6). HaPYL2a has the highest number of interacting proteins at 19 and the abscisic acid 8′-hydroxylase CYP707A2 only interacts with HaPYL2a, suggesting that HaPYL2a may perform more roles in the sunflower than other HaPYLs.

The tissue expression pattern of the HaPYL gene family revealed that multiple HaPYLs were detected in developing sunflower tissue (Fig. 7). This result aligns with the findings in many plants such as tobacco (Bai et al., 2019), cucumber (Zhang et al., 2022), rice (Yadav et al., 2020), cotton (Zhang et al., 2017), and maize (Fan et al., 2016), which suggests that plant development requires the coordinated functioning of multiple PYL genes. The coordinated and cooperative work among PYL genes may facilitate new functionalization during antagonistic coevolution in response to complex and changing environments (Yang et al., 2020). In A. thaliana, PYL9 promoted ABA-induced leaf senescence. Senescence and death of old leaves increased after the overexpression of AtPYL9 but the young tissues survived under severely limited water conditions by promoting summer dormancy-like responses (Asad et al., 2019). As the orthologues of PYL9 in A. thaliana, HaPYL9a and HaPYL9b, which are found mainly in leaves, may play a similar role in the sunflower. HaPYL8a is consistently highly expressed at early stages of flower development (St2-St4) in the sunflower and is expressed at much higher levels than that of the other HaPYL genes. HaPYL8a may play a special role in flower initiation and polarization in the sunflower. Previous studies on the function of ABA signaling during the plant development process mainly focused on fruit ripening, seed dormancy and germination, lateral root development, and regulation of the leaf stomatal opening. Fewer studies have been conducted on its function in flower development and controversy still exists regarding the contribution of ABA to floral transition (Conti, Galbiati & Tonelli, 2014; Martignago et al., 2020; Shu et al., 2018). Future studies on the function of HaPYL8a in the early stages of flower development in the sunflower may provide new insight into the effects of ABA on floral transition. ABA played an important role in embryo development and seed maturation in tobacco and rice, and the ABA receptors (PYLs) were essential ABA signaling components that function mainly in seeds (Frey et al., 2004; Miao et al., 2018). HaPYL8b and HaPYL8c have the highest expression levels in sunflower seeds at the developmental and maturation stages (st7-st8). HaPYL8b and HaPYL8c are tightly clustered in the phylogenetic tree with NtPYL21, NtPYL22, NtPYL26, and NtPYL27 found in tobacco. Given that these NtPYLs were expressed at high levels in dry seeds and cotyledons, HaPYL8b and HaPYL8c may play an important role in regulating the development and maturation of the sunflower seed.

Gene expression is regulated by diverse developmental and environmental signals and transcriptional initiation is an important part of this regulation. The analysis of cis-regulatory elements showed that multiple responses were detected in the promoter regions of the 19 HaPYL genes, mainly in abiotic stress response-related elements, hormone-related elements, and other elements that regulate plant growth and development (Fig. 8). These cis-regulatory elements are the same as those detected in other species, indicating that PYL genes have similar transcriptional response patterns across plants (Lu et al., 2020; Yadav et al., 2020; Zhang et al., 2022; Zhou et al., 2023). PYL genes may facilitate responses to low temperature, anaerobic conditions, and drought stress through their low-temperature response, anaerobic induction, and drought induction components. A variety of phytohormone-responsive elements, including abscisic acid, auxin, gibberellin, salicylic acid, and methyl jasmonate, were found in the HaPYL promoter sequences. This suggests that in response to developmental and environmental stimuli, HaPYL proteins not only act as direct abscisic acid receptors but also interact with components of other phytohormone signaling pathways and may even be directly involved in the regulation of downstream stress-related proteins.

ABA is an important stress hormone in plants. When plants are exposed to abiotic stress such as drought and high salt conditions, ABA levels in tissues increase, which in turn activates the ABA signaling and triggers the abiotic stress resistance response of a plant (Saddhe, Kundan & Padmanabh, 2017; Sah, Reddy & Li, 2016). Most HaPYL genes respond to exogenous application of ABA and have different response patterns, which may indicate the different functions they serve in abiotic stress resistance in the sunflower. To understand the response of HaPYLs to abiotic stress, their altered expression levels was examined using qRT-PCR under treatment with 10% PEG6000 and 100 mM NaCl. The results showed that 12 HaPYLs were up-regulated under treatment with 10% PEG6000, including HaPYL8a, HaPYL8b, HaPYL8c, HaPYL9b, HaPYL9c, HaPYL4d, HaPYL4g, HaPYR1b, HaPYR2a, HaPYR2b, HaPYR2c, and HaPYR2d (Fig. 9). Among these genes, HaPYL8a, HaPYL8b, HaPYL8c, HaPYL9b, and HaPYL9c were tightly clustered with ZmPYL8, ZmPYL9, ZmPYL12 in the phylogenetic tree (Fig. 1). A previous study indicated that the overexpression of ZmPYL8, ZmPYL9, and ZmPYL12 promoted drought tolerance in maize (He et al., 2018). Moreover, HaPYR1b, HaPYR2a, HaPYR2b, HaPYR2c, and HaPYR2d were homologous to OsPYL3 found in rice and the drought tolerance of rice was improved by OsPYL3 overexpression (Lenka et al., 2018). Sarazin et al. (2017), revealed that the PYL4 gene promoted drought tolerance of the sunflower, as this gene had higher expression levels in the least drought-sensitive line under drought conditions compared with normal drought-sensitive lines. Based on the function of HaPYL homologous genes under drought conditions in previous reports, we speculate that these 12 HaPYL genes could play an important role in drought tolerance in the sunflower.

Among the 12 HaPYL genes responsive to drought stress, HaPYL8a, HaPYL8b, HaPYL8c, HaPYL9b, HaPYL9c, HaPYL4d, HaPYL4g, and HaPYR2a were up-regulated by NaCl-treatment (Fig. 9). The results suggest that these eight genes not only play a role in drought tolerance but also in salt resistance of the sunflower. Nevertheless, other HaPYLs showed altered expression levels that were significantly different between PEG6000 and NaCl treatments, e.g., HaPYL9a and HaPYL2b, which were up-regulated under PEG6000 treatment but down-regulated under NaCl treatment. These results suggest that a shared signal pathway mediated by common ABA receptors exists in the drought and salinity response of the sunflower. Some unique pathways could drive the sunflower’s resistance to drought and salinity stress by using specific ABA receptors. Interestingly, the eight genes that responded to drought as well as salinity stress possessed the motif 5 and/or motif 6, which were close to START domain in location. The results imply that motif 5 and motif 6 might function in mediating the ABA signaling via ABA perception, thus contributing to the sunflowers’ ability to respond to abiotic stress (such as drought and salinity stress).

Conclusions

In higher plants, ABA signaling is an important pathway for increasing the plant’s tolerance to abiotic stress, such as drought and salt stress. Under drought and salt stress, an induced dehydration effect would trigger the hyperosmotic signaling and cause abscisic acid (ABA) biosynthesis, which actives the ABA signaling associated with resistance to drought and salt stress in plants. In this process, PYLs play a crucial role in mediating the transduction of ABA signaling when related to drought and salt stress resistance. Nineteen PYL genes were identified from the sunflower. Among these genes, eight HaPYLs (HaPYL8a, HaPYL8b, HaPYL8c, HaPYL9b, HaPYL9c, HaPYL4d, HaPYL4g, and HaPYR2a) responded intensively to drought and salt stress. Our results suggest that these eight HaPYLs could play a role in the sunflower’s resistance to drought and salt stress. Perhaps there exists a common ABA signal pathway, mediated by HaPYL receptors, that regulates the drought and salt resistance of the sunflower. In addition, these eight genes contained motif 5 and/or motif 6, which might play an important role in ABA perception, and influence ABA signal pathway related to plant’s response to drought and salt stress.

Supplemental Information

Supplemental Information 1 Supplemental tables

Supplemental Information 2 Raw data of qRT-PCR

Additional Information and Declarations

Competing Interests

Author Contributions

Data Availability

The authors declare there are no competing interests.

Zhaoping Wang conceived and designed the experiments, performed the experiments, analyzed the data, prepared figures and/or tables, authored or reviewed drafts of the article, and approved the final draft.

Jiayan Zhou conceived and designed the experiments, performed the experiments, analyzed the data, prepared figures and/or tables, authored or reviewed drafts of the article, and approved the final draft.

Jian Zou conceived and designed the experiments, analyzed the data, authored or reviewed drafts of the article, and approved the final draft.

Jun Yang conceived and designed the experiments, analyzed the data, authored or reviewed drafts of the article, and approved the final draft.

Weiying Chen conceived and designed the experiments, analyzed the data, authored or reviewed drafts of the article, and approved the final draft.

The following information was supplied regarding data availability:

The raw measurements are available in the Supplementary File.

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
