# Peer review of "Characterization of PYL gene family and identification of HaPYL genes response to drought and salt stress in sunflower"

_PeerJ, doi:10.7717/peerj.16831_

## Round 0.1 · original submission · Major Revisions

The authors are requested to revise the manuscript as per the suggestions of the reviewers. Also requested the authors to kindly check spelling and English grammar carefully in the manuscript. References are needed to re-look.

**Language Note:** The Academic Editor has identified that the English language must be improved. PeerJ can provide language editing services - please contact us at copyediting@peerj.com for pricing (be sure to provide your manuscript number and title). Alternatively, you should make your own arrangements to improve the language quality and provide details in your response letter. – PeerJ Staff

·

Basic reporting

It would be helpful to provide more context on the significance of drought as an abiotic stress affecting agricultural production and the importance of developing drought-resistant crops. Additionally, a brief overview of the abscisic acid (ABA) signaling pathway and its role in plant responses to abiotic stresses could enhance the understanding of the study's objectives.

Experimental design

The study mentions the identification of 19 members of the PYL family in sunflower, but it would be beneficial to describe the methodology used for this identification, such as genome mining, transcriptome analysis, or other techniques. Providing details on the criteria used for gene selection and the specific databases or tools employed would enhance the reproducibility of the study.

While the study mentions the division of HaPYL genes into three subfamilies based on phylogenetic analysis, it would be informative to include a phylogenetic tree or a visual representation to illustrate the relationships among the genes. This would allow readers to better visualize the clustering and evolutionary relationships of the PYL genes in sunflower.

Validity of the findings

The study mentions the expression analysis of HaPYL genes in different tissues and under various abiotic stress treatments. To provide a comprehensive understanding, it would be helpful to include the expression patterns of the identified HaPYL genes across different tissues and compare their expression levels under different stress conditions. Additionally, including statistical analysis or replicates for the expression data would strengthen the reliability of the findings.

The study proposes several potential regulatory mechanisms of HaPYL proteins for drought resistance based on predicted protein interactions. It would be beneficial to elaborate on these mechanisms and provide supporting evidence or references for the predicted interactions. Additionally, experimental validation or functional studies would greatly enhance the credibility of these proposed mechanisms.

Additional comments

The study mentions that the results provide a foundation for further elucidation of the function of the PYL gene family in sunflowers and analysis of its regulatory mechanisms under drought conditions. It would be valuable to outline specific future research directions or hypotheses that could build upon the current findings and contribute to a deeper understanding of the role of PYL genes in sunflower drought tolerance.

Reviewer 2 ·

Basic reporting

no comment

Experimental design

no comment

Validity of the findings

My decision is thus a significant major revision that should permit more confidently evaluate the data provided.
1. Please follow the journal guidelines throughout the manuscript. Sometimes gene names are mentioned in italics and sometimes not. And the protein names should not mentioned in italics. Please follow the same pattern everywhere.
2. The expression of HaPYLs under abiotic stresses was examined, including PEG6000, NaCl, and ABA treatments. Why the title of the manuscript is present just in osmotic stress?
3. For the same reason in 2, the descriptions of salt stress (not just drought or osmotic stress) should be added in the Abstract, Keywords, Introduction and other places in the manuscript.
4. What’s the difference of drought and osmotic stress? Both of these two descriptions were occurs in the manuscript.
5. The authors need to write in detail the importance and role of sunflower in human life and economy. They should make the connection why this species needs to be studied and what will be the importance of PYL genes in sunflower.
6. Author needs to cite the proper references for plant PYL genes and that is the key for this paper.
7. Does the orientation and pattern of occurrence of motifs show any correlation with the stress factors?
8. The figures are relevant for the content of the article, however most of them are rough and lack of information, and the resolution should be improved.
9. Statistical analysis is missed for qRT-PCR.
10. In the manuscript some grammatical mistakes can be found, please correct them.

Additional comments

no comment

---

## Round 0.2 · Major Revisions

The authors are again requested to revise the manuscript as per the suggestions of the reviewer 2. Also requested the authors to kindly check spelling and English grammar carefully in the manuscript.

**Language Note:** The Academic Editor has identified that the English language must be improved. PeerJ can provide language editing services - please contact us at copyediting@peerj.com for pricing (be sure to provide your manuscript number and title). Alternatively, you should make your own arrangements to improve the language quality and provide details in your response letter. – PeerJ Staff

·

Basic reporting

The authors have revised the manuscript as per the instructions. I am happy that now the manuscript is acceptable in its current form.

Experimental design

NA

Validity of the findings

NA

Additional comments

NA

Reviewer 2 ·

Basic reporting

no comment

Experimental design

Statistical analysis in the qRT-PCR results of many genes are incorrect.

Validity of the findings

Thanks for the author's serious response for previous comments. However, there were still several problems into consideration for revision.
1. The expression of HaPYLs under different abiotic stresses was examined in this study, including PEG6000, NaCl, and ABA treatments. Why the title of the manuscript is present just in osmotic stress? I think that the descriptions of PEG6000, NaCl, and ABA stress should be added in the title.
2. What’s the difference of drought and osmotic stress? Both of these two descriptions were occurs in the manuscript. I think that osmotic stress is different from the drought and salt stress. The authors should noted that and descript the difference of them in the manuscript.
3. Statistical analysis results of many genes are incorrect in the qRT-PCR results, such as HaPYL8c, HaPYL4d, HaPYR1a, HaPYR1b, HaPYL2b, HaPYL2c and HaPYL2d in PEG6000, as well as some genes in salt and ABA qRT-PCR results.

---

## Round 0.3 · Minor Revisions

The authors are requested to revise the manuscript as per the suggested points carefully.

**Language Note:** The review process has identified that the English language must be improved. PeerJ can provide language editing services - please contact us at copyediting@peerj.com for pricing (be sure to provide your manuscript number and title). Alternatively, you should make your own arrangements to improve the language quality and provide details in your response letter. – PeerJ Staff

Reviewer 2 ·

Basic reporting

no comment

Experimental design

no comment

Validity of the findings

The manuscript has some undeserved problems that need careful revision. It is suggested that the revised version be published after checking the grammatical and spelling mistakes in the manuscript.

Additional comments

no comment

---

## Round 0.4 · accepted · Accept

The manuscript can be accepted in its current form.